# Utility of Three-Dimensional Cultures of Primary Human Hepatocytes (Spheroids) as Pharmacokinetic Models

**DOI:** 10.3390/biomedicines8100374

**Published:** 2020-09-23

**Authors:** Kenta Mizoi, Hiroshi Arakawa, Kentaro Yano, Satoshi Koyama, Hajime Kojima, Takuo Ogihara

**Affiliations:** 1Faculty of Pharmacy, Takasaki University of Health and Welfare, 60, Nakaorui-machi, Takasaki, Gunma 370-0033, Japan; mizoi-k@takasaki-u.ac.jp (K.M.); arakawa@p.kanazawa-u.ac.jp (H.A.); kentarou.yano@hamayaku.ac.jp (K.Y.); koyamasatoshi.3@gmail.com (S.K.); 2Faculty of Pharmacy, Institute of Medical, Pharmaceutical and Health Sciences, Kanazawa University, Kakuma-machi, Kanazawa, Ishikawa 920-1192, Japan; 3Faculty of Pharmacy, Yokohama University of Pharmacy, 601, Matano-cho, Totsuka-ku, Yokohama, Kanagawa 245-0066, Japan; 4Center for Biological Safety and Research, National Institute of Health Sciences, 3-25-26, Tono-machi, Kawasaki-ku, Kawasaki, Kanagawa 210-9501, Japan; h-kojima@nihs.go.jp; 5Graduate School of Pharmaceutical Sciences, Takasaki University of Health and Welfare, 60, Nakaorui-machi, Takasaki, Gunma 370-0033, Japan

**Keywords:** spheroids, 3D culture, primary human hepatocyte, pharmacokinetics, metabolism, toxicity, CYP induction, drug-induced liver injury

## Abstract

This paper reviews the usefulness, current status, and potential of primary human hepatocytes (PHHs) in three-dimensional (3D) cultures, also known as spheroids, in the field of pharmacokinetics (PK). Predicting PK and toxicity means pharmaceutical research can be conducted more efficiently. Various in vitro test systems using human hepatocytes have been proposed as tools to detect hepatic toxicity at an early stage in the drug development process. However, such evaluation requires long-term, low-level exposure to the test compound, and conventional screening systems such as PHHs in planar (2D) culture, in which the cells can only survive for a few days, are unsuitable for this purpose. In contrast, spheroids consisting of PHH are reported to retain the functional characteristics of human liver for at least 35 days. Here, we introduce a fundamental PK and toxicity assessment model of PHH spheroids and describe their applications for assessing species-specific metabolism, enzyme induction, and toxicity, focusing on our own work in these areas. The studies outlined in this paper may provide important information for pharmaceutical companies to reduce termination of development of drug candidates.

## 1. Introduction

This review addresses in vitro evaluation of pharmacokinetics (PK) and associated toxicity, a crucial step in pharmaceutical research. In particular, this paper introduces a fundamental pharmacokinetic and toxicity assessment model of three-dimensional (3D) culture of primary human hepatocytes (PHHs), i.e., spheroids. Since predicting PK and toxicity means pharmaceutical research can be conducted more efficiently, these findings may prove advantageous to pharmaceutical companies.

In drug development, ratios of progression/termination calculated from 812 oral small molecule drug candidates at AstraZeneca, Eli Lilly and Company, GlaxoSmithKline, and Pfizer were 0.17, 0.45, and 0.61 for candidate nomination, Phase I, and Phase II, respectively [1]. Recent estimates indicate that the average cost of developing a new drug ranges from $314 million to $2.8 billion [2]. In addition, total capitalized costs of new drug development have been reported to increase at an annual rate of 8.5% [3]. During drug discovery and the preclinical stages of drug development, it is necessary to investigate the efficacy, PK, and toxicity of drug candidates using a variety of methods. These can be broadly classified into in vitro methods using materials derived from cells and tissues and in vivo methods using laboratory animals. As regards PK, it is important to study the absorption, distribution, metabolism, and excretion of candidate drugs in the liver, small intestine, and kidney [4]. In this context, hepatocytes and liver-derived materials are widely used in studies such as metabolite analysis (i.e., metabolomics) [5,6], inhibition or induction of cytochrome P450 (CYP) enzymes (at the mRNA, protein, and activity levels) [7,8,9], non-CYP enzyme metabolism [10,11], and prodrug metabolism [12,13]. On the other hand, hepatotoxicity (i.e., drug-induced liver injury, DILI) is often evaluated as part of an exploratory safety screening in the evaluation of new drug candidates. However, despite such screening, DILI remains one the main reasons for the abandonment of drug development and for withdrawal of approved drugs from the market [1,14,15,16]. In other words, serious DILI is sometimes only detected during clinical trials or even after marketing. Furthermore, DILI is difficult to diagnose clinically [17]. Therefore, there is still a need for better systems to predict hepatotoxicity at an early stage.

Experimental animals or animal-derived cells are usually used in toxicity studies, but unfortunately the metabolic enzymes and metabolic pathways of drugs differ fundamentally between humans and experimental animals [18]. In fact, half of the drugs found to cause hepatic dysfunction during clinical trials did not produce liver damage in experimental animals [19,20]. Much research experience has shown that animal models can be useful, but the validity of extrapolating the results obtained from animal models to predict human risk remains problematic, mainly because of the lack of suitable preclinical and clinical datasets [21].

Therefore, various in vitro test systems using human hepatocytes have been proposed as tools to detect hepatic toxicity early in the drug screening process [22,23]. In particular, PHHs have been reported to be the most accurate cell model for the prediction of hepatotoxicity [24,25]. However, even in cultured PHHs, the expression levels of many metabolic enzymes gradually decrease during the culture process, and the resulting decrease of metabolic capacity has led to misinterpretation in toxicity evaluation [26,27]. In other words, DILI assessment requires long-term, low-level exposure to the test compound, but conventional screening systems such as PHHs in planar (2D) culture can only survive for a few days. Indeed, 2D cultures have other important limitations. For example, 2D-cultured cells do not mimic the natural tissue structures. In addition, cell–cell and cell–extracellular environment interactions are not reproduced, and these interactions are involved in cell differentiation, proliferation, drug metabolism, and other cellular functions. In addition, cells in 2D culture have essentially unlimited access to medium components such as oxygen, nutrients, metabolites, and signaling molecules, in marked contrast to the in vivo situation [28].

To overcome these drawbacks, 3D-cultured hepatocyte systems, which can maintain functions specific to the human liver for a long period of time, have been developed [29]. Various methods using spheroids, hollow fiber bioreactors, and organ-on-chip systems have been proposed (Table 1) [30]. In particular, the advantages of spheroids (3D aggregates of cells) compared to these systems are long-term stability and versatility. Moreover, spheroids are relatively inexpensive, require minimal technological investment, and are compatible with high-throughput screening (HTS). On the other hand, the disadvantage of spheroids is that they are difficult to handle. Spheroids are considered to improve the relevance of in vitro results to the in vivo situation and to provide better biological models of native tissues. For example, in concert with other cellular aggregates, they can develop complex tissue architectures. Further, cells in spheroid cultures secrete an extracellular matrix in which they reside, and they can interact with cells from their original microenvironment [31]. Spheroids can also be formed from human cancer-derived hepatocytes, such as HepaRG cells [32,33,34], HepG2 cells [35,36,37], and Huh-7 cells [38]. Cultured human cancer-derived hepatocytes have long been used for human-specific toxicity assessments, but they generally have lower expression levels of drug-metabolizing enzymes than PHHs [39,40]. Various applications of PHH spheroids as pharmacokinetic models [41,42,43,44] or hepatotoxicity detection models [16,45,46,47,48,49,50,51] have already been reported, and it has become clear that this model is extremely versatile and offers a variety of advantages over 2D cultures (Table 2) [16,41,42,43,44,45,46,47,48,49,50,51,52,53,54,55,56,57]. PHH spheroids have also been used as disease models [58,59,60].

An initial search of papers containing the keywords “spheroid” and “3D model” garnered more than 2000 candidate results. Adding “liver”, “in vitro” and “human” to the search reduced this number to 140 papers. Further refinement via the keyword “primary” reduced that number to approximately 50. Furthermore, once the cell line was removed from these results, we found just 15 papers focusing on pharmacokinetic and toxicity studies using PHHs. In the following sections, we review the current status of applications of spheroid cultures of PHHs to PK (see Table 2), including our own studies using PHH spheroids to assess drug metabolism, enzyme induction, and associated toxicity. We also introduce related evaluation methods using hepatocytes isolated from liver-humanized mice, hepatocytes differentiated from liver progenitor cells, and hepatocytes derived from human induced pluripotent stem (iPS) cells, which may have similar capabilities to PHHs.

## 2. Recent 3D-Culture Studies of PHHs for Pharmacokinetic Models

Hepatic drug metabolism plays a key role in drug PK in humans, and various pharmacokinetic models have been developed for evaluating candidate drugs. For example, Vorrink et al. reported that the metabolic activity of CYP1A2, CYP2C8, CYP2C9, CYP2D6, and CYP3A4 is more stable in 3D spheroid cultures than in 2D monolayer cultures [41]. Specifically, the functional activities of these enzymes were decreased by >95% after 7 days in 2D culture, while in 3D spheroids they were stable for 21 days (Figure 1). Xia et al. compared CYP induction in 3D spheroid culture and 2D sandwich culture [42,43] and found that CYP3A4 mRNA induction was significantly greater in the spheroid culture. The activities of CYP1A2, CYP2B6, and CYP3A4 were similar in the two culture systems. Desai et al. reported a CYP induction/inhibition evaluation system using a 3D cell culture prepared by aggregation of PHHs with magnetic nanoparticles using magnets placed under the wells [44]. This system could evaluate the induction/inhibition of CYP1A2, CYP2B6, and CYP3A4.

## 3. Recent 3D-Culture Studies Using PHHs for Hepatotoxicity Detection Models

PHH spheroids are useful as hepatotoxicity detection models because of their long-term viability compared with 2D cultures. Vorrink et al. reported an assessment of DILI for 123 compounds using PHH spheroids [16]. With ATP quantification as a single endpoint, this spheroid model was able to accurately distinguish between hepatotoxic and non-toxic structural analogs. It detected 69% of hepatotoxic compounds at low exposure levels, and no false positives were found among known non-hepatotoxic compounds (Figure 2). Bell et al. compared 3D spheroid and 2D sandwich cultures over a long period [45,46] and found that the enzyme activities of CYP1A2, CYP2C8, CYP2C9, CYP2D6, and CYP3A4 were significantly higher in spheroid culture than in sandwich culture. The obtained EC_50_ values of hepatotoxic drugs such as acetaminophen, bosentan, diclofenac, fialuridine, and troglitazone also indicated that spheroid culture is more sensitive to long-term exposure. In this context, it has been reported that the hepatic phenotype of 3D-cultured spheroids is well maintained during long-term culture (for at least 35 days), as evidenced by well-sustained ATP levels, albumin secretion, and the overall stability of CYP enzyme activities [46]. Moreover, Bell et al. evaluated acetaminophen-induced hepatotoxicity using a 3D co-culture system of PHHs and non-parenchymal cells, noting that high-level expression of miRNAs involved in the toxicity, i.e., mi-382 and mi-155, was observed [47]. In addition, to evaluate the potential of their system as a predictive model for drug-induced hepatotoxicity, they compared three emerging cell systems: hepatocytes derived from induced pluripotent stem cells, HepaRG cells, and PHH spheroids, at the transcriptional and functional levels in a multicenter study [48]. Transcriptome analysis revealed important phenotypic differences among the three cell systems, and the results indicated that PHH spheroids were suitable to investigate DILI. Parmentier et al. studied individual differences in response to cholestatic hepatotoxins using PHHs obtained from different donors in 2D sandwich or 3D spheroid culture [49]. They found that there was an interpersonal difference in susceptibility to drug-induced cholestasis, which was compound- and exposure-time-dependent. Interestingly, the sensitivity of PHHs to cholestasis toxicity of several compounds increased with prolonged exposure in 3D spheroid cultures. Li et al. created co-culture spheroids of PHHs and Kupffer cells. Elsewhere, a study comparing PHH spheroids with co-cultured spheroids for evaluation of 14 types of DILI-causing compounds showed that the role of Kupffer cells was compound-dependent [50]. Hendriks et al. evaluated two 3D spheroid models, from PHHs and HepaRG, to detect cholestatic compounds [51]. Marked selective synergistic toxicity due to a mixture of compounds known to cause cholestatic/hepatocellular toxicity and non-toxic concentrated bile acids was observed when the exposure time was extended to 14 days. In contrast, no synergistic effect of a mixture of compounds causing non-cholestatic hepatotoxicity and concentrated bile acids was observed at 8 or 14 days.

## 4. Application of PHH Spheroids for Analysis of Drug Metabolism

In 2014, we reported the usefulness of PHH spheroids for metabolite analysis [52], using 96-well plates processed for spheroid formation (Cell-able^®^, Toyo Gosei, Tokyo, Japan). First, supporter (feeder) cells to promote hepatocyte adhesion were seeded and cultured for 4 days, and then PHHs were added. The hepatocytes formed spheroids, which maintained the cellular morphology for at least 21 days after seeding. The mRNA expression levels of multiple drug-metabolizing enzymes (CYP1A2, CYP2C9, CYP2C19, CYP2D6, CYP2E1, CYP3A4, UGT1A1, UGT2B7, SULT1A1, and GSTP1) at 2 and 7 days after seeding were similar to those of uncultured fresh hepatocytes. The spheroids were exposed to acetaminophen, diclofenac, lamotrigine, midazolam, propranolol, and salbutamol from 2 days after seeding, and metabolites were analyzed after 2 and 7 days of drug exposure by means of triple quadrupole liquid chromatography mass spectrometry (LC-MS/MS), with reference to our earlier report [61]. Continuous metabolism of midazolam, diclofenac, and propranolol by phase I and phase II enzymes was observed. In the case of midazolam, 1′- and 4-hydroxymidazolam were detected on day 2 of exposure. These primary metabolites decreased or disappeared by day 7, and the corresponding glucuronate conjugates were detected as secondary metabolites. Notably, the 4-hydroxylated metabolites were difficult to detect using conventional methods. In the case of diclofenac, 4′- and 5-hydroxydiclofenac were detected as primary metabolites on day 2. Subsequently, acylglucuronate conjugates and 4′-hydroxyacylglucuronate conjugates were observed (Figure 3), whereas these were difficult to detect using conventional human hepatocyte suspension methods [62]. Although it is believed that glucuronide conjugates are generally non-toxic, the acylglucuronates of diclofenac are hepatotoxic [63,64,65]. On the other hand, lamotrigine and salbutamol were metabolized to lamotrigine-*N*-glucuronate and salbutamol 4-*O*-sulfate conjugate, respectively. These metabolites, which are human-specific, were not observed in a conventional liver culture system [66]. Such metabolites can be traced semi-quantitatively by using LC-MS/MS, even if authentic synthetic samples are not available, because they often generate the same fragment ions as the corresponding intact drugs. These data show that culture systems using PHH spheroids offer many advantages over conventional methods for tracing sequential human-specific metabolic processes from phase I to phase II. Various culture plates can be used in this analysis, from 12- to 384-well, and either clear wall or black wall, depending on the purpose of the test.

## 5. Application of PHH Spheroids for Prediction of Drug-Induced Liver Injury

We have also examined the usefulness of PHH spheroids for the evaluation of DILI [53]. Thus, spheroids were exposed to 11 known hepatotoxins (acetaminophen, benzbromarone, chlorpromazine, cyclosporin A, diclofenac, fialuridine, flutamide, imipramine, isoniazid, ticlopidine, and troglitazone) from 2 days after hepatocyte seeding. When hepatotoxicity is evaluated in vitro, cell viability and ATP production are generally used as indicators. However, these indicators reflect cell death, and thus are not necessarily suitable as indexes for toxicity assessment in the case of relatively long-term exposure to low concentrations of test compounds. Therefore, for prediction of DILI, we chose leakage of aspartate aminotransferase (AST), albumin secretion, and changes of cell morphology as indexes. Clinically, AST leakage is increased in liver damage, and AST is used as a cellular injury marker [67]. We used the concentration of test compound that increases AST leakage to 120% of that in the unexposed state (F_1.2_ value of AST) as a criterion of “increased AST”. For albumin secretion, the 50% inhibitory concentration (IC_50_ value) of the test compound was used as an index. When the above compounds were evaluated under these conditions, the IC_50_ value of albumin and the F_1.2_ value of AST changed between day 7 and day 14, but remained stable between day 14 and day 21, except in the case of diclofenac. Therefore, a 14-day drug exposure period was considered to be sufficient to assess hepatotoxicity. For many of the compounds tested, there was a positive correlation between the IC_50_ value for albumin secretion and the F_1.2_ value for AST leakage, supporting their availability as indicators of drug-induced hepatotoxicity (Figure 4). On the other hand, this was not the case for cyclosporin A or fialuridine. The reason for this might be that these drugs induce mitochondrial dysfunction, and AST is specifically localized in mitochondria, so AST leakage is particularly high. A comparison of the IC_50_ values obtained in this work with those found in cytotoxicity assays using cultured PHHs and HepG2 cells (Table 3) suggested that the conventional evaluation methods may underestimate hepatotoxicity. Notably, a dramatic change in the IC_50_ of fialuridine from 18.1 to 0.9 µM was observed between day 7 and day 21 after exposure, and the latter value is close to the clinically relevant plasma concentration of fialuridine [68]. Thus, the availability of PHH spheroids for comparatively long-term toxicity assessment is clearly advantageous, in this case at least.

## 6. Effect of Feeder Cells on Application of PHH Spheroids for Prediction of Hepatotoxicity

In 3D-culture methods, feeder cells are often used to improve hepatocyte adhesion, but these feeder cells also have a protective effect on the hepatocytes. In addition, toxic compounds can alter the enzyme activities of feeder cells, so only albumin secretion is available as a parameter for tracing toxicity. Therefore, we devised a system without feeder cells using Matrigel^®^ Basement Membrane Matrix (CORNING, Corning, NY, USA) [54]. For 14 days, spheroids cultured with or without feeder cells (Sph(f+))/(Sph(f−)) were exposed to the following hepatotoxic drugs: flutamide, diclofenac, isoniazid, and chlorpromazine, at different concentrations. As toxicity markers, AST, alanine aminotransferase (ALT), lactate dehydrogenase (LDH), and γ-glutamyl transpeptidase (γ-GTP) were measured. No consistent difference in ALT leakage was found between Sph(f+) and Sph(f−), but leakage in AST, LDH, or γ-GTP from Sph(f−) was similar to or less than that from Sph(f+) across all tested drugs. Specific to Sph(f−), notable correlations were detected among all toxicity markers aside from γ-GTP. The IC_50_ value for albumin secretion from Sph(f−) and the F_1.2_ value for AST leakage were equal to or less than those from Sph(f+) for all tested drugs. These results confirm that feeder cells have a protective effect on hepatocytes, suggesting that DILI can be more accurately assessed using by means of the Sph(f−) system.

## 7. Application of PHH Spheroids for Evaluating Induction of Drug-Metabolizing Enzymes

In general, metabolic reactions are thought to contribute to the detoxification of drugs. However, since DILI can be induced by metabolites, liver toxicity might be increased in the enzyme-induced state in some cases. The Organisation for Economic Co-operation and Development (OECD), an international organization that aims to provide internationally agreed methodology for assessing the safety of chemicals, has recently proposed a draft test guideline for assessing CYP-inducing activity [9]. This guideline recommends the use of 2D-cultured HepaRG cells [69]. However, as already discussed, this method is not suitable for long-term exposure studies. Therefore, we investigated whether the induction of drug-metabolizing enzymes could be evaluated using PHH spheroids, and we compared the results with those from 2D culture [55].

CYP induction is activated by the associated xenobiotic receptors and transcription factors; e.g., CYP1A2, CYP2B6, and CYP3A4 are induced by the activation of aryl hydrocarbon receptor (AhR, *AHR*), constitutive androstane receptor (CAR, *NR1I3*), and pregnane X receptor (PXR, *NR1I2*), respectively [70]. Because inclusion of mRNA expression assays of these genes is preferable as part of a CYP induction study, we examined the basal mRNA expression levels of AhR, CAR, and PXR in non-cultured, 2D-cultured, and 3D-cultured hepatocytes. The expression level of AhR mRNA in 3D-cultured hepatocytes tended to be higher than in non-cultured hepatocytes (i.e., hepatocytes before seeding in culture plates), whereas the expression levels of CAR and PXR mRNAs in 3D-cultured hepatocytes tended to be lower than in non-cultured hepatocytes. Expression levels of these genes in 3D-cultured hepatocytes were stable for at least l4 days after seeding, while those in 2D culture were unstable between days 2 and 7. Therefore, 3D-cultured hepatocytes appear to be more suitable for evaluating the induction of metabolic enzymes than 2D-cultured hepatocytes. Two days after exposure of PHH spheroids to omeprazole (OPZ), phenobarbital, and rifampicin, typical inducers of CYP1A2, CYP2B6, and CYP3A4, respectively, the mRNA expression levels of these CYPs were significantly higher than the control levels in the absence of inducers. We found that the level of CYP2B6 mRNA was stable between days 2 and 14 of exposure, while the levels of both CYPlA2 and CYP3A4 continued to increase until day l4. The fold inductions of all the examined mRNAs were lower in 2D culture than in 3D culture. Furthermore, in the 3D culture, when these enzyme activities were followed using a standard substrate, the induction factor of the enzyme activity was also significantly increased until day 14 compared to the control (Figure 5). The metabolic activities were well correlated with the mRNA expression levels. Overall, these results suggest that the PHH spheroid system is preferable for long-term study of the induction of enzyme activities.

## 8. Application of PHH Spheroids for Evaluating Metabolic Toxicity under Conditions of Enzyme Induction

DILI may be enhanced by drug–drug interactions because when CYP is induced, the quantity of metabolites of any concomitantly administered drugs may be increased, presenting an increased risk of DILI if the metabolites and/or the metabolic process are toxic [71,72,73,74]. Therefore, it is important to assess the ability of compounds to cause DILI in the presence of CYP inducers. We developed a CYP1A2-mediated metabolic toxicity system using PHH spheroids treated with OPZ as a CYP1A2 inducer and dacarbazine (DTIC) as a test substrate [56], since CYP1A2 catalyzes the formation of a cytotoxic product (reactive oxygen species) from DTIC [75]. Cell viability, CYP1A2 mRNA expression level, and DTIC metabolism were measured alongside hepatic function markers (albumin and urea secretion and AST leakage).

Fold induction of CYP1A2 mRNA was examined 16 days after seeding, (nine days after OPZ addition), and it was found to be notably increased in an OPZ concentration-dependent manner. Eleven and 16 days after seeding (two and seven days after DTIC addition), amounts of unchanged DTIC and produced metabolite (AIC) in the supernatant were investigated. Without the addition of OPZ, AIC was detected in the supernatants, and approximately 50% of DTIC was metabolized after two days and about 75% after seven days. The amount of produced AIC was significantly increased and the amount of unchanged DTIC was significantly decreased when 30 μM OPZ was added. Moreover, hepatic function markers were measured on day 16 after seeding. These hepatic function markers were significantly decreased by the addition of OPZ and DTIC (Figure 6). However, single addition of OPZ and DTIC or simultaneous addition of the two had almost no effect on the viability of the spheroids. We consider that this system detected CYP1A2-mediated metabolic toxicity with high sensitivity, and so it should be useful for evaluation of the metabolic toxicity of compounds and their potential to cause DILI in the presence of CYP1A2 inducers.

## 9. Culture of PHH Spheroids on NanoCulture Plates

Devices that maintain mechanical adhesion without the use of feeder cells are commercially available; an example is the NanoCulture Plates (NCPs, Medical & Biological Laboratories, Aichi, Japan), a novel microstructural plate designed as a base for the 3D culture of cells/tissues. If spheroids become too large, sufficient oxygen does not reach the cells in the interior, and these cells may die. We have shown that PHH spheroids constructed on NCPs retain an acceptable size with sufficient gaps to remain viable [57].

Post-seeding, the hepatocytes formed aggregates and maintained consistent ATP content for their 21 days of culture. Throughout the 21-day culture period, expression of CYP1A2, CYP2B6, CYP2C9, CYP2C19, CYP2D6, CYP2E1, and CYP3A4 mRNAs was detected. The addition of acetaminophen, diclofenac, lamotrigine, and midazolam (CYP substrate drugs) led to the formation of numerous metabolites and a corresponding decrease in the quantity of unchanged compounds. 1′-hydroxymidazolam, which was observed on day 2, was notably not detected on day 7. 4′-Hydroxydiclofenac acyl glucuronide was formed by hepatocytes on an NCP, whereas in previous studies, diclofenac acyl glucuronide had not been detected in human hepatocyte suspensions [62,66]. These results indicate that PHH spheroids on NCPs could prove a useful model for sequential drug metabolism evaluation via UGT and CYP. Moreover, despite the non-detection of the NAPQI metabolite in our previous system [52], it was found using an NCP, suggesting that NCPs may be of value in the examination of hepatotoxic metabolite formation during long-term exposure and for assessing chronic toxicity. Lamotrigine N2-glucuronide, the main metabolite of lamotrigine in humans, was also detected. Typical CYP inducers work well on PHH spheroids on NCPs. OPZ, phenobarbital and rifampicin increased the mRNA expression levels of CYP1A2, CYP2B6, and CYP3A4 by 110-fold, 12.5-fold, and 5.4-fold, respectively, at 2 days after inducer treatment. The CYP activities were also increased (CYP1A2, 2.2-fold; CYP2B6, 20.6-fold; CYP3A4, 3.3-fold). Therefore, the NCP should be useful as a substrate for PHH spheroids for a variety of research purposes, such as assessing the chronic toxicity of new compounds, assessing CYP induction, identifying metabolites, validating drug targets, and developing disease models.

## 10. Other Hepatocyte Culture Systems as Potential Alternatives to PHHs

The supply of PHHs is limited, so it is necessary to use cultured PHHs, even though differences among lots in terms of metabolic enzyme activity and survival rate will inevitably occur. In order to overcome this issue, the use of hepatocytes isolated from liver-humanized mice, mature hepatocytes differentiated from liver progenitor cells, or hepatocytes derived from human iPS cells instead of PHHs has been investigated.

In the case of hepatocytes isolated from liver-humanized mice, human hepatocytes are transplanted into mice with both immunodeficiency and hepatic failure. This generates a mouse model having a liver with the characteristics of human hepatocytes. When the liver is isolated and cultured, the original cells of the mouse disappear, and cells with characteristics similar to those of human hepatocytes are formed. These cells can be obtained in larger quantities than conventional cultured PHHs, and the engraftment success rate is high, so the number of samples that can be processed in the same lot is greatly increased [76]. In addition, because the cells inherit the properties of the transplanted donor hepatocytes, this system may have potential utility for personalized medicine. The hepatocytes isolated from liver-humanized mice have been used in a wide variety of applications, including studies on the life cycle of the hepatitis B virus [77,78]. Recently, hepatocytes isolated from liver-humanized mice (such as PXB-cells^®^ (PhoenixBio, Hiroshima, Japan)) have been suggested to be useful for assessing human liver function. For example, Watari et al. cultured hepatocytes isolated from liver-humanized mice using collagen vitrigel membrane and evaluated their CYP activity, albumin secretion, and urea synthesis [79]. Similarly, Kanamori et al. performed a metabolic test of butyrylfentanil, taking advantage of the high phase I and phase II enzymatic activities of hepatocytes isolated from liver-humanized mice, and examined the metabolites and CYPs involved in the production of these metabolites [80]. In the case of mature hepatocytes differentiated from liver progenitors, rat or mouse hepatocytes are converted to chemically induced liver progenitors (CLiPs) using a cocktail of small molecules. CLiPs can differentiate into both hepatocytes and biliary epithelial cells. Long-term cultured CLiPs did not lose their ability to proliferate and differentiate into mature hepatocytes [81]. Although there has so far been little work on spheroids derived from hepatocytes isolated from liver-humanized mice or differentiated from liver progenitors, these areas are likely to receive increasing attention in the future.

On the other hand, it has been suggested that if hepatocytes with functions equivalent to those of PHHs could be created from human iPS cells, which can grow indefinitely, they could be a valuable alternative to PHHs [82]. Yamashita et al. reported a method for large-scale production of homogeneous and functional hepatocyte-like cells from human iPS cells [83]. Moreover, Sirenko et al. performed 2D culture and 3D spheroid culture experiments using human iPS cell-derived hepatocytes and conducted toxicity tests using various hepatotoxic drugs [84]. Some drugs showed slightly lower IC_50_ values in 3D cultures, suggesting the spheroids are more sensitive for detecting toxicity.

Thus, there is a range of possible approaches to increase the versatility of hepatocyte spheroids, and further developments can be anticipated.

## 11. Conclusions

PHH spheroids have some drawbacks in HTS application (see Table 1) and may not be useful when this application is used for early screening in new drug development. However, many drugs have had to be withdrawn due to the occurrence of hepatotoxicity immediately prior to or soon after launch. To avoid this problem, hepatocytes need to be exposed to low concentrations of the drug for relatively long periods to accurately observe toxicity. PHH spheroids can meet that demand. For example, values obtained from PHH spheroids better reflect clinical toxicity, while those obtained from cell lines may have been underestimated (see Table 3). The studies outlined in this paper provide important information to aid pharmaceutical companies in reducing the termination of drug candidate development, and PHH spheroids may prove to be of value when narrowing down candidate compounds for new drug development.

The usefulness of the 3D-cultured PHH spheroid model for assessing species-specific metabolism, enzyme induction, and associated toxicity is already well established. Based on our long-term exposure studies with PHH spheroids, we speculate that in vitro assessment of liver function involves three stages, as shown in Figure 7. Firstly, hepatocyte function decreases with concomitant loss of liver function markers. We think that this corresponds to “reduction of liver function markers under conditions that do not affect cell viability” in metabolic toxicity study (see Section 7). Secondly, damage to hepatocytes accumulates, and functional markers are secreted, as represented by “F_1.2_ values of AST leakage” in DILI studies (see Section 4). Thirdly, the induction of cell death results in a rapid loss of functions. Many in vitro toxicity studies have “cell death” as an endpoint, which may occur at this stage.

We believe spheroids are powerful in vitro tools for predicting the pharmacokinetic and toxicological properties of human drug candidates and can contribute greatly to drug development. Spheroids are not restricted to PHHs and can be formed using various cells, devices, or methods. Therefore, they should be available for a wide range of research applications, and they offer significant advantages over 2D culture models. We hope that this review of the current status of PHH spheroids will encourage their broader application.

## Figures and Tables

**Figure 1 biomedicines-08-00374-f001:**
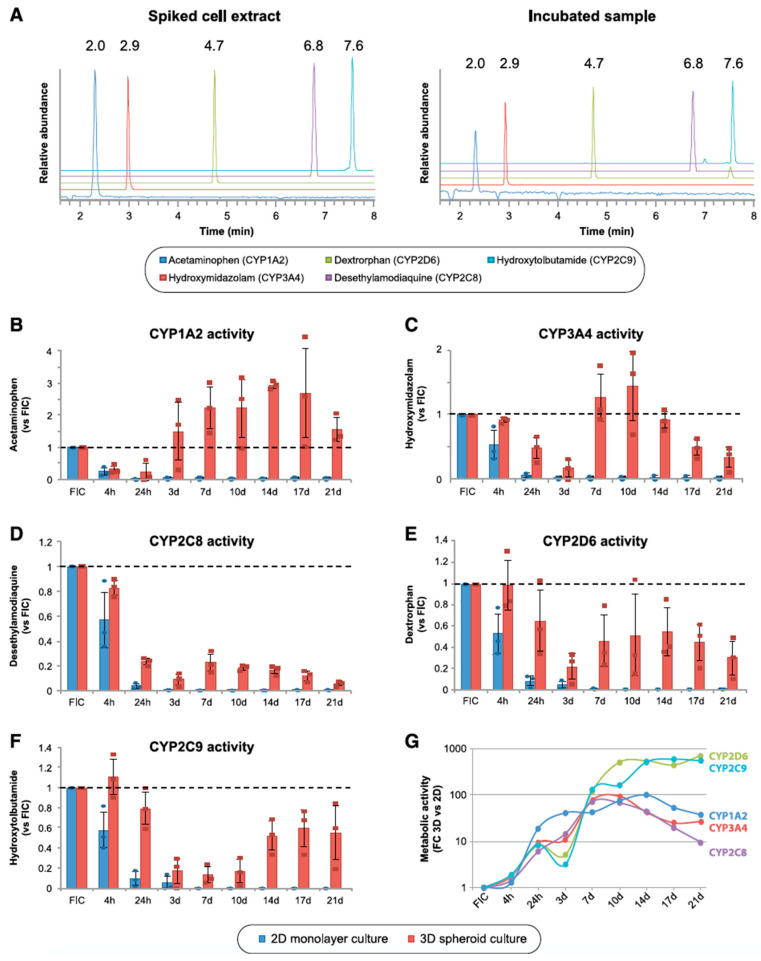
PHHs in 3D spheroid culture show significantly higher functional activities of major human cytochrome P450 (CYP) enzymes compared with PHHs in 2D culture. (**A**) Chromatograms of primary metabolites (acetaminophen, hydroxymidazolam, desethylamodiaquine, dextrorphan, and hydroxytolbutamide) of five CYP probe substrates in a spiked cell extract (each 8 ng/mL) and in a sample incubated for 4 h. Numbers above the peaks are retention times. (**B**–**F**) The time courses of the metabolic activities of CYP1A2 (**B**), CYP3A4 (**C**), CYP2C8 (**D**), CYP2D6 (**E**), and CYP2C9 (**F**) in PHHs from three donors cultured in 2D monolayer or 3D spheroid cultures. Dashed lines: metabolite levels in freshly isolated cells (FICs), taken as unity. Error bars = SD. (**G**) Line plot of changes of CYP activity ratios in 3D vs. 2D culture in panels (**B**–**F**) (FC: fold change). CYP metabolic activities are much greater in the spheroids during prolonged incubation. Reprinted from reference [41].

**Figure 2 biomedicines-08-00374-f002:**
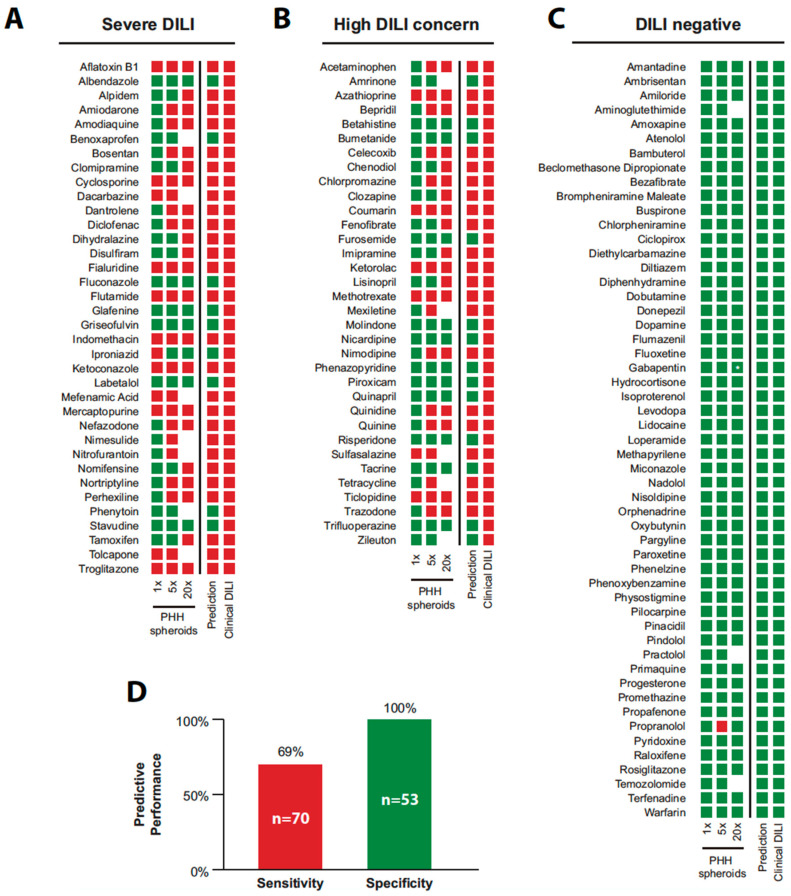
Compounds with potential hepatotoxicity indicated by chemically defined-spheroid model. Results of drug-induced liver injury (DILI) evaluation are shown for those with acute DILI (*n* = 36; (**A**)), with potential DILI (*n* = 34; (**B**)), and DILI-negative (*n* = 53; (**C**)). Red boxes show the average hepatocyte viability to be significantly decreased (<80% of the control, *p* < 0.05). The rating is classified as negative (green) when these conditions are not met. (**D**) The spheroid model indicated 69% of DILI-positive drugs as hepatotoxic with no false positives. Reprinted from reference [16].

**Figure 3 biomedicines-08-00374-f003:**
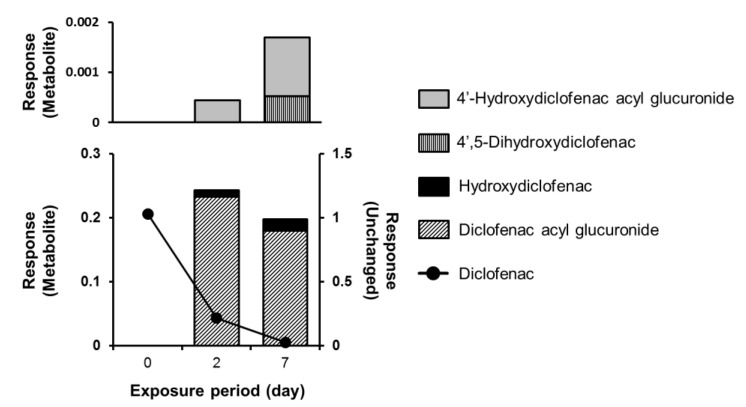
Time-dependent metabolism of diclofenac by PHH spheroids. The spheroids were exposed to 10 μM diclofenac for 2 or 7 days after a spheroid formation period of 2 days. Closed circles represent unchanged diclofenac. The metabolites were identified by comparing the retention times and molecular weights with those of the standard compounds. The quantity of metabolites was determined as the relative ratio to the ion peak area of an internal standard. The experiment was performed in duplicate.

**Figure 4 biomedicines-08-00374-f004:**
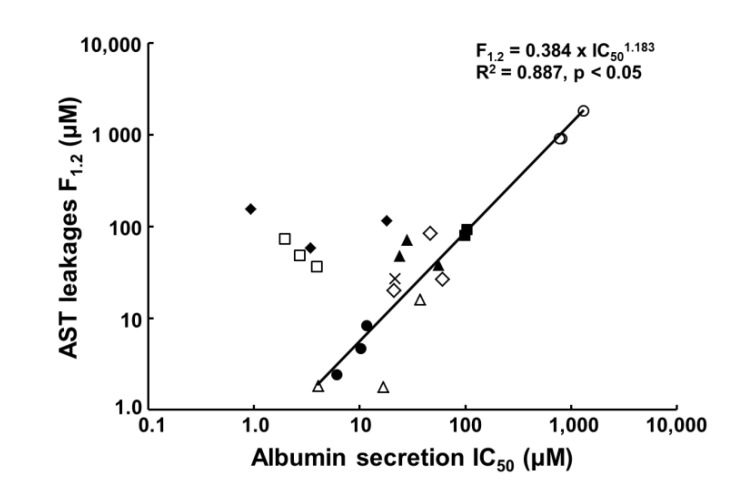
Correlation between albumin secretion (IC_50_) and aspartate aminotransferase (AST) leakage (F_1.2_) in toxicity evaluation of compounds with PHH spheroids. The IC_50_ values and F_1.2_ values of acetaminophen (open circles), chlorpromazine (closed circles), cyclosporine A (open squares), diclofenac (closed squares), fialuridine (closed rhombuses), flutamide (open rhombuses), imipramine (open triangles), ticlopidine (closed triangles), and troglitazone (cross marks) are shown. There is a strong positive correlation, except for cyclosporine A and fialuridine. Reprinted from reference [53].

**Figure 5 biomedicines-08-00374-f005:**
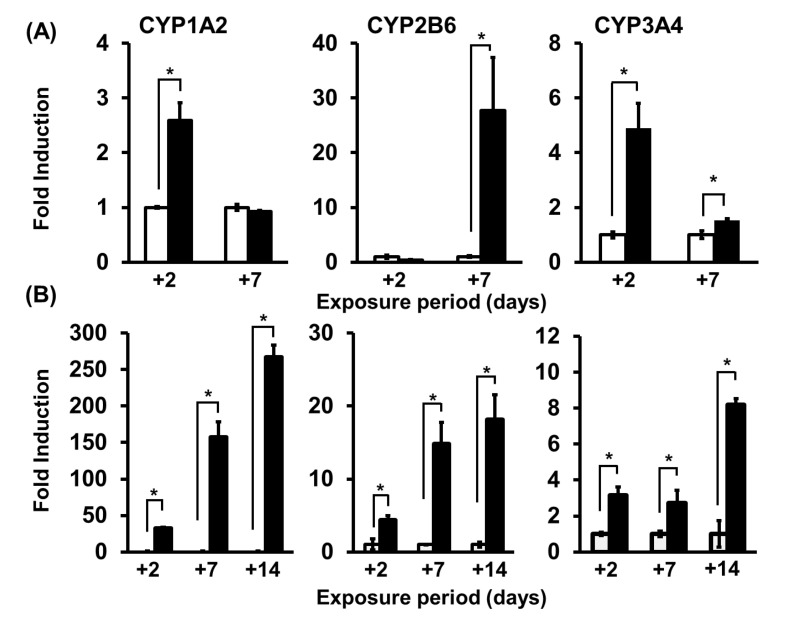
Induction of enzymatic activity of CYP1A2, CYP2B6, and CYP3A4/5 in 2D- and 3D-cultured hepatocytes. (**A**) 2D culture. Two days after seeding, human hepatocytes were exposed to inducers; 100 μM of omeprazole for CYP1A2, 1 mM of phenobarbital for CYP2B6, or 10 μM of rifampicin for CYP3A4/5. DMSO (0.1%) was used as vehicle control. After exposure to test compounds for 2 and 7 days, enzyme activity was measured (phenacetin O-demethylation to form acetaminophen, bupropion hydroxylation to form hydroxybupropion, and midazolam hydroxylation to form 1′-hydroxymidazolam). (**B**) On day 7, after formation of 3D structures, hepatocytes were exposed to inducers, and enzyme activities were measured as above. Open and closed columns represent the activity of each CYP enzyme when cells were exposed to DMSO (0.1%) or inducers for 2, 7, and 14 days. The columns indicate the mean ± SEM (*n* = 3–4) of fold induction versus the control. * significant differences from the control (*p* < 0.05). Reproduced with permission from Biol. Pharm. Bull. Vol. 40 No. 7. pp 967–974 [55]. Copyright 2017 The Pharmaceutical Society of Japan.

**Figure 6 biomedicines-08-00374-f006:**
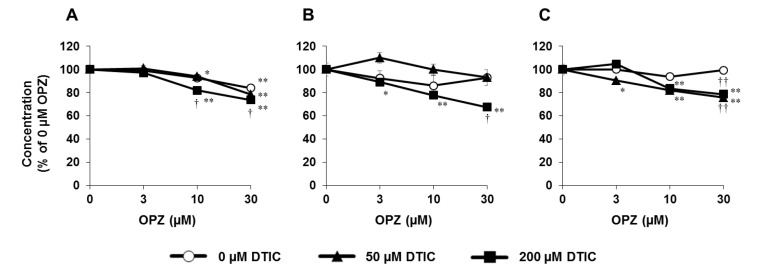
Metabolic toxicity assay using PHH spheroids. The 3D spheroids were exposed to omeprazole (OPZ) (0, 3, 10, or 30 µM) and dacarbazine (DTIC) (0, 50, or 200 µM). Albumin secretion (**A**), urea secretion (**B**), and AST leakage (**C**) were measured. Values are mean ± S.E. (*n* = 3). * *p* < 0.05, ** *p* < 0.01 compared with 0 µM OPZ. † *p* < 0.05, †† *p* < 0.01 compared with 0 µM DTIC. Quoted with permission from Drug Metab. Pharmacokinet. Vol. 35 No. 2. pp 201–206 [56]. Copyright 2019, The Japanese Society for the Study of Xenobiotics.

**Figure 7 biomedicines-08-00374-f007:**
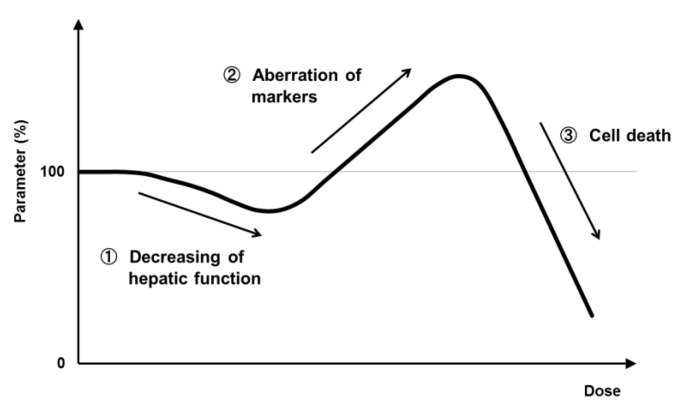
Hypothesis regarding the different stages (endpoints) of in vitro hepatotoxicity.

**Table 1 biomedicines-08-00374-t001:** Comparison of hepatocyte culture methods.

	Long-Term Stability	High-Throughput Capability	Versatility	Required Cell Numbers	Complexity	Scaffold
2D monolayer	−	●●●	−	●●	●	Collagen
Sandwich culture	●● (2 weeks)	●	−	●●	●●	Collagen or Matrigel
3D spheroids	●●● (>5 weeks)	●●	●●●	●	●●	Mostly scaffold-free
Hollow-fiber bioreactors	●●● (>5 weeks)	−	●	●●●	●●●	Synthetic polymers
Micro-patterned co-cultures	●●● (>5 weeks)	●●	●●	●●	●●	Collagen-coated islands
Perfused multiwell plates	●● (2 weeks)	●●	●●	●●	●●	Extracellular matrix -coated polymer wafer
Microfluidic liver biochips	●● (1–4 weeks)	●●	●●	●●	●●●	Mainly scaffold-free
Microfluidic multiorgan devices	●●● (>4 weeks)	−	●●	●●	●●●	Mainly scaffold-free

●●●, ●●, and ● indicate high, medium, and low, respectively. Adapted with permission from Chem. Res. Toxicol. Vol. 29 No. 12. pp 1936–1955 [30]. Copyright 2016 American Chemical Society.

**Table 2 biomedicines-08-00374-t002:** Summary of experiments using primary human hepatocyte (PHH) spheroid cultures.

	Category	Purpose of Experiment	Plate/Membrane	Culture Period	Reference
	Pharmacokinetic models				
1	Metabolic activity of CYPs	Examination of metabolic stability using spheroid and 2D monolayer cultures	Ultra-low attachment plate (CORNING)	Culture for 21 days after seeding	[41]
2	Induction of CYPs	Comparison of spheroid and collagen sandwich cultures	Arginine, glycine, and aspartic acid/galactose-conjugated membrane	Culture for 5 days after seeding Induction for 2 days from 3 days after seeding	[42]
3	Induction of CYPs	Evaluation of the usefulness of spheroid cultures (compared with sandwich cultures)	Arginine, glycine, and aspartic acid/galactose-conjugated membrane	Culture for 5 days after seeding Induction for 2 days from 3 days after seeding	[43]
4	Induction/inhibition of CYPs	Assembly and handling of magnetic 3D cell culture	Cell-repellent plate (CELLSTAR, Greiner Bio-One)	Drug exposure for 3 days from 3 days after seeding	[44]
	Hepatotoxicity detection models				
5	Hepatotoxicity	Evaluation of hepatotoxicity of 123 drugs using spheroid cultures	Ultra-low attachment plate (CORNING)	Drug exposure for 14 days from 7 days after seeding	[16]
3	Hepatotoxicity Metabolic activity of CYPs Proteomics	Comparison of spheroid and 2D sandwich cultures at six laboratories	Ultra-low attachment plate (CORNING)	Drug exposure for 14 days from 7–10 days after seeding	[45]
4	Hepatotoxicity Metabolic activity of CYPs Proteomics	Evaluation of the usefulness of spheroid cultures	Ultra-low attachment plate (CORNING)	Culture for 35 days after seeding	[46]
5	Hepatotoxicity	Applied a 3D co-culture system of acetaminophen-induced toxicity	Ultra-low attachment plate (CORNING)	Drug exposure for 14 days from 8 days after seeding	[47]
6	Hepatotoxicity Transcriptomics	Compared three emerging cell systems at transcriptional and functional levels in a multicenter study	Ultra-low attachment plate (CORNING)	Drug exposure for 14 days from 7 days after seeding	[48]
7	Hepatotoxicity	Assess the inter-donor variability in the response of PHHs towards cholestatic compounds	Ultra-low attachment plate (CORNING)	Drug exposure for 14 days from 8 days after seeding	[49]
8	Hepatotoxicity	To evaluate the role of Kupffer cells in DILI using co-culture spheroids	Spheroid microplate (CORNING)	Culture for 15 days after seeding	[50]
9	Hepatotoxicity	Evaluation of two 3D spheroid models for the detection of compounds with cholestatic liability	Ultra-low attachment plate (CORNING)	Drug exposure for 14 days from 5–6 or 5 or 8 days after seeding	[51]
	Our studies				
10	Drug metabolism	Metabolic experiments using spheroid cultures	Micro-patterned plate (Cell-able, Toyo Gosei)	Culture for 21 days after seeding Drug exposure for 2 or 7 days	[52]
11	Hepatotoxicity	Utility of spheroid for evaluation of hepatotoxicity	Micro-patterned plate (Cell-able, Toyo Gosei)	Drug exposure for 21 days from 2 days after seeding	[53]
12	Hepatotoxicity	Evaluation of hepatotoxicity using spheroid cultures (with or without feeder cells)	Micro-patterned plate (Cell-able, Toyo Gosei)	Drug exposure for 14 days from 2 days after seeding	[54]
13	Induction of CYPs	Metabolic induction experiment using spheroid cultures (compared with 2D cultures)	Micro-patterned plate (Cell-able, Toyo Gosei)	Induction for 14 days from 7 days after seeding	[55]
14	Induction of CYP1A2 Drug metabolism Metabolic toxicity	Evaluation of metabolic toxicity using spheroid cultures	Micro-patterned plate (Cell-able, Toyo Gosei)	Culture for 16 days after seeding Drug exposure for 7 days	[56]
15	Induction of CYPs Drug metabolism	Evaluation of the usefulness of spheroid cultures using NanoCulture Plate	Micro-patterned plate (NanoCulture plate, MBL)	Culture for 21 days after seeding	[57]

**Table 3 biomedicines-08-00374-t003:** Comparison of IC_50_ values for albumin secretion between PHH spheroids and conventional assays.

Compound.	Albumin Secretion IC_50_ (μM)			Reported IC_50_ (μM) of Conventional Assays	Clinical C_max_ (μM)
	Day 7	Day 14	Day 21		
Acetaminophen	1295.2	809.3	772.4	28,200 (HH) 29,755 (HepG2)	139
Benzbromarone	48.8	<20	22.2	>40 (HepG2)	4.3
Chlorpromazine	10.3	11.7	6.1	1.73–18.3 (HH) 42.9–62.6 (HepG2)	1.41
Cyclosporine A	3.9	2.7	2.0	24.4–56.8 (HH) >100 (HepG2)	0.78
Diclofenac	98.4	103.3	104.6	331 (HH) 763 (HepG2)	8.1
Fialuridine	18.1	3.4	0.9	>400 (HepG2)	0.64
Flutamide	21.0	60.4	46.5	6.29–100 (HH) >100 (HepG2)	4.16
Imipramine	37.0	4.1	16.8	37 (HepG2)	0.14
Isoniazid	>1000	254.1	336.2	>10,000 (HepG2)	76.6
Ticlopidine	55.8	23.9	28.1	Not reported	7.1
Troglitazone	42.0	46.6	21.5	>50 (HH) 30 (HepG2)	6.4

HH: 2D culture of PHHs. Modified from reference [53].

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
