# Peer review of "Utility of Three-Dimensional Cultures of Primary Human Hepatocytes (Spheroids) as Pharmacokinetic Models"

_biomedicines, 2020, doi:10.3390/biomedicines8100374_

Round 1
Reviewer 1 Report
Additions and corrections are sufficient for acceptance.
Author Response
Reviewer 1
Additions and corrections are sufficient for acceptance.
[Reply]
Thank you for your interest in our paper.
Reviewer 2 Report
In the revised version the authors have added two small parts with pictures from two additional publications. However the manuscript still does not at all represent an informative and critical review of the field. All previous criticism raised in the original version still essentially remains. Thus, there is still a complete lack of critical evaluation of the different hepatic 3D model systems mentioning what the applications can be, how such systems are used their properties, advantages and disadvantages and their ability to simulate the situation in vivo in human liver. Additionally human liver spheroids have been used for many other applications than for those presented by the authors
Author Response
Reviewer 2
In the revised version the authors have added two small parts with pictures from two additional publications. However the manuscript still does not at all represent an informative and critical review of the field. All previous criticism raised in the original version still essentially remains. Thus, there is still a complete lack of critical evaluation of the different hepatic 3D model systems mentioning what the applications can be, how such systems are used their properties, advantages and disadvantages and their ability to simulate the situation in vivo in human liver. Additionally human liver spheroids have been used for many other applications than for those presented by the authors
[Reply]
Thank you for your advice. The types of 3D cultured hepatocyte systems were summarized in Table 1. In order to emphasize the characteristics of spheroids, we have added and modified the explanation as shown in lines 83–93 of section 1 as follows: “Various methods using spheroids, hollow fiber bioreactors, and organ-on-chip systems have been proposed (Table 1) [30]. In particular, the advantages of spheroids (3D aggregates of cells) compared to these systems are long-term stability and versatility. Moreover, spheroids are relatively inexpensive, require minimal technological investment, and are compatible with high-throughput screening (HTS). On the other hand, the disadvantage of spheroids is that they are difficult to handle. Spheroids are considered to improve the relevance of in vitro results to the in vivo situation, and to provide better biological models of native tissues. For example, in concert with other cellular aggregates, they can develop complex tissue architectures. Further, cells in spheroid cultures secrete an extracellular matrix in which they reside, and they can interact with cells from their original microenvironment [31].”
In addition, we emphasized in lines 444–453 of section 10 that our study reflects in vivo as follows: “PHH spheroids have some drawbacks in HTS application (see Table 1) and may not be useful when this application is used for early screening in new drug development. However, many drugs have had to be withdrawn due to the occurrence of hepatotoxicity immediately prior to or soon after launch. To avoid this problem, hepatocytes need to be exposed to low concentrations of the drug for relatively long periods to accurately observe toxicity. PHH spheroids can meet that demand. For example, values obtained from PHH spheroids better reflect clinical toxicity, while those obtained from cell lines may have been underestimated (see Table 3). The studies outlined in this paper provide important information to aid pharmaceutical companies in reducing the termination of drug candidate development, and PHH spheroids may prove to be of value when narrowing down candidate compounds for new drug development.”
As you pointed out, human liver spheroids are used for many purposes. In fact, there are many papers when expanding to spheroid research using animal-derived cells or human-derived cell lines. Those applications also have many reports. However, as mentioned in the introduction of this paper, we believe it is important to utilize primary human hepatocytes (PHH). Our keywords are "spheroids" and "PHH". So, we narrowed down the number of articles for our paper. Few articles used PHH in their reporting of spheroids. There were still fewer reports featuring pharmacokinetic and toxicity models. Among current papers, we believe our research to be of value to pharmaceutical companies. We have therefore added the following to lines 100–107 of section 1 regarding the narrowing down process that led us the reports referenced in this paper. “An initial search of papers containing the keywords “spheroid” and “3D model” garnered more than 2,000 candidate results. Adding “liver”, “in vitro” and “human” to the search reduced this number to 140 papers. Further refinement via the keyword "primary" reduced that number to approximately 50. Furthermore, once the cell line was removed from these results, we found just 15 papers focusing on pharmacokinetic and toxicity studies using PHH. In the following sections, we will review the current status of applications of spheroid cultures of PHH to pharmacokinetics (see Table 2), including own our studies using PHH spheroids to assess drug metabolism, enzyme induction, and associated toxicity.”
Round 2
Reviewer 2 Report
All the criticism previously raised regarding the previous versions remain. The additions in this revised version are not sufficient in order to classify this contribution as an expert opinion. It is written in an uncritical fashion highlighting some parts of this important field but fails to encompass work done by other groups than the author's group with one exception.
Author Response
All the criticism previously raised regarding the previous versions remain. The additions in this revised version are not sufficient in order to classify this contribution as an expert opinion. It is written in an uncritical fashion highlighting some parts of this important field but fails to encompass work done by other groups than the author's group with one exception.
[Reply]
Thank you for your input. While these comments are appreciated, we believe the specific area covered in this paper does not warrant the criticism. We focused this paper on studies using primary human hepatocyte spheroids for pharmacokinetic and toxicity assessments. These studies may provide important information for new drug development. However, if, as you suggest, we have overlooked relevant work by others, we would welcome a more detailed response, including specific examples appropriate for this report.
This manuscript is a resubmission of an earlier submission. The following is a list of the peer review reports and author responses from that submission.
Round 1
Reviewer 1 Report
This review of the use of hepatic spheroids is useful and timely. I expect it will be a helpful primer for individuals interested in a review of both the basic history of spheroid technology and its application, however upon further review, it is a too much self-referencing and the work requires significant additional sources and data references. Minor grammatical mistakes were sprinkled throughout the manuscript can be corrected in short order.
Author Response
This review of the use of hepatic spheroids is useful and timely. I expect it will be a helpful primer for individuals interested in a review of both the basic history of spheroid technology and its application, however upon further review, it is a too much self-referencing and the work requires significant additional sources and data references. Minor grammatical mistakes were sprinkled throughout the manuscript can be corrected in short order.
[Reply]
In accordance with your suggestions, we have expanded the review to include the results of other researchers. We have made corrections throughout the manuscript, so please refer to the highlighted copy in the text. Moreover, the revised manuscript has been grammar checked again by an experienced, native-speaking scientific editor. Thank you for pointing that out.
Reviewer 2 Report
This manuscript is aimed at reviewing the usefulness, current status and potential of primary human hepatocytes (PHH) in three-dimensional (3D) cultures.
The contribution is limited to the 3D spheroid system and in particular to the data generated by the authors themselves, indeed all 8 figures (With possible exception from fig 2, where the origin is not given) are copy-pasted from the authors own publications (!). In addition the tables (at least table 3) are copy-pasted from other papers by the authors. So almost all of the data/results presented have previously been published by the authors.
The area as such, namely liver spheroids gives 1,124 hits in PubMed, and they are used for many different applications. In a review on this topic, at least work from other labs than that from the authors should be discussed.
There is a complete lack of critical evaluation of the different hepatic 3D model systems as mentioning what the applications can be and how such systems are used.
Human liver spheroids have been used for many other applications than for the pharmacokinetic analyses presented by the authors.
The title is confusing, “Utility of three-dimensional cultures of primary hepatocytes (spheroids) as models of human liver”. What is meant? Models for liver function, liver diseases etc?
Figure 2 has no legend. And from where is it taken?
Author Response
This manuscript is aimed at reviewing the usefulness, current status and potential of primary human hepatocytes (PHH) in three-dimensional (3D) cultures.
The contribution is limited to the 3D spheroid system and in particular to the data generated by the authors themselves, indeed all 8 figures (With possible exception from fig 2, where the origin is not given) are copy-pasted from the authors own publications (!). In addition the tables (at least table 3) are copy-pasted from other papers by the authors. So almost all of the data/results presented have previously been published by the authors.
The area as such, namely liver spheroids gives 1,124 hits in PubMed, and they are used for many different applications. In a review on this topic, at least work from other labs than that from the authors should be discussed.
There is a complete lack of critical evaluation of the different hepatic 3D model systems as mentioning what the applications can be and how such systems are used.
Human liver spheroids have been used for many other applications than for the pharmacokinetic analyses presented by the authors.
The title is confusing, “Utility of three-dimensional cultures of primary hepatocytes (spheroids) as models of human liver”. What is meant? Models for liver function, liver diseases etc?
[Reply]
Thank you for your valuable suggestions. The purpose of this paper is to review the model of human liver in 3D culture of primary hepatocytes (spheroids), focusing on pharmacokinetic studies and its associated toxicity assessment. The title has been changed to “Utility of three-dimensional cultures of primary human hepatocytes (spheroids) as pharmacokinetic models”. In fact, when we researched papers involving "liver" and "3D model", more than 700 candidate papers were found. When these were refined with the terms “Vitro” and “Human”, it became 170 papers. Further reading of these contents narrowed down the number of papers that specifically describe “human cells” to 60. In addition, once the cell line was removed, pharmacokinetic and toxicity studies using primary hepatocytes were reported in 15 papers. We have rewritten this review including these papers. We have made corrections throughout the manuscript, so please refer to the highlighted copy in the text. Thank you for pointing this out again.
Figure 2 has no legend. And from where is it taken?
[Reply]
Thank you for your suggestion. This data was obtained from Reference [50]. We wrote the legend in Figure 2 (Figure 3 in the current manuscript) as follows: “The spheroids were exposed to 10 μM diclofenac for 2 or 7 days after a spheroid formation period of 2 days. Closed circles represent unchanged diclofenac. The metabolites were identified by comparing the retention times and molecular weights with those of the standard compounds. The amounts of metabolites were determined as the relative ratio to the ion peak area of an internal standard. The experiment was performed in duplicate.”